materials science/chemical engineering

graphene oxide, network structure, drug delivery

**Authors for correspondence:**
Sunnam Kim
e-mail: sn-kim@kumamoto-u.ac.jp
Seiji Kurihara
e-mail: kurihara@gpo.kumamoto-u.ac.jp

This article has been edited by the Royal Society of Chemistry, including the commissioning, peer review process and editorial aspects up to the point of acceptance.

# Adsorption and release on three-dimensional graphene oxide network structures

Sunnam Kim, Sho Moriya, Sakura Maruki, Tuyoshi Fukaminato, Tomonari Ogata and Seiji Kurihara

Department of Materials Science and Applied Chemistry, Kumamoto University, 2-39-1 Kurokami, Chuo-ku, Kumamoto 860-8555, Japan

SK, 0000-0001-5039-7569

In this study, three-dimensional network architectures are constructed using nano-sized graphene oxide (nGO) as the building block. The cross-linking reaction of nGO is conducted in sub-micrometre water droplets in an emulsion system to control the size of the networks by restricting the reaction space. Two types of three-dimensional GO networks with different cross-linking lengths were constructed, and their methyl orange adsorption and release behaviours were investigated under external stimuli, such as thermal treatment, ultrasonic wave treatment and near-infrared light irradiation.

## 1. Introduction

The development of new and effective drug delivery systems (DDSs) for therapeutic management is one of the most important challenges of modern medicine. Graphene, comprising a single layer of $sp^2$-hybridized carbon atoms arranged in a hexagonal lattice, has attracted significant interest throughout the scientific community. Due to its unique structure and geometry, graphene possesses an excellent capability to immobilize many substances, including metals, drugs and biomolecules, as well as fluorescent probes and cells. Namely, a suitably modified graphene material can serve as an excellent drug delivery platform for anti-cancer/gene delivery, bio-sensing, bio-imaging, antibacterial applications, and cell culture and tissue engineering [1–8]. The theoretical specific surface area of graphene ($2630\,m^2\,g^{-1}$) is larger than that of any other nanomaterial explored for drug delivery [9]. Basically, a graphene monolayer represents an extreme case, where every atom on the surface is exposed, which accounts for its significantly higher drug loading capacity compared with other nanomaterials. However, pristine graphene is highly hydrophobic and exhibits poor dispersibility in water; consequently, it requires surfactants or surface modification to be viable for any biological application.

On the other hand, graphene oxide (GO), generally produced by the chemical exfoliation of oxidized graphite, possesses abundant oxygen functional groups, such as hydroxyl and epoxide groups on its basal plane, and carboxyl groups at its edges, which promote high dispersibility in pure water and also provide reactive sites for further surface chemical functionalization. These excellent properties of aqueous processability, amphiphilicity and surface functionalizability make GO excellent for use in DDSs [10–16].

GOs are conventionally polydisperse materials, with sizes ranging from a few nanometres to tens of micrometres. However, since the size and geometry of a drug delivery vehicle are essential for overcoming sequential biological barriers, the control of the size of GO architectures for bio-applications is instrumental. Moreover, GOs exhibit a strong tendency to form irreversible agglomerates of multilayers in solutions rich in salts or proteins, such as cell media and sera, through strong π–π stacking and van der Waals interactions [17–19]. Therefore, to use a GO-based delivery system, the size and its distribution of GO must be controllable, and the aggregation must be suppressed.

Here, to solve these problems, we propose a three-dimensional network structure with GO as the building block. The three-dimensional GO network structures were prepared by cross-linking nano-sized GOs (nGOs), and the cross-linking reaction sites were limited by nano-sized water droplets in a water-in-oil (w/o) emulsion system [20]. The target size of three-dimensional GO networks was in the range of 10–500 nm, and this structure is very suitable for DDS application [21–23]. The methyl orange adsorption and release behaviours on the novel nano-sized three-dimensional GO networks with different cross-link lengths were investigated under external stimuli, such as thermal treatment, ultrasonic wave treatment and near-infrared light irradiation.

# 2. Experimental

## 2.1. Materials

By the modified Hummers method, nGO was prepared using platelet graphite nanofibres (PGNFs) as the starting material [13]. The PGNF (diameter: 50 nm) was purchased from Strem Chemicals, Inc. Potassium peroxo-disulfate and phosphorus pentoxide were purchased from Nacalai Tesque, Inc. Sorbitan monooleate (Span 80), ethylenediamine (EDA) and 2, 2′-oxybisethylamine (OBEA) were purchased from Tokyo Chemical Industry Co., Ltd. Methyl orange was purchased from FUJIFILM Wako Pure Chemical Corp. The purchased chemicals were used without further purification.

## 2.2. Characterization

Structural analysis was performed by X-ray photoelectron spectroscopy (XPS), X-ray diffraction (XRD, Rigaku Rint 2500 HV, 40 kV, 200 mA), Fourier transform infrared spectroscopy (FTIR, PerkinElmer Co., Ltd., Frontier), atomic force microscopy (AFM, Agilent 5500), transmission electron microscopy (TEM, JEM-14000Plus, 100 kV), high-resolution transmission electron microscopy (HRTEM, Tecnai F20, 200 kV) and dynamic light scattering (DLS, Malvern ZETASIZER Nano series).

## 2.3. Synthesis of nGO

In a round-bottom flask, 0.15 g of potassium peroxo-disulfate and 0.15 g of phosphorus pentoxide were added to 5 ml of sulfuric acid ($H_2SO_4$), after which the mixture was heated to 80°C. Subsequently, 0.2 g of PGNF was added, followed by stirring at 80°C for 4.5 h. Thereafter, the mixture was cooled to room temperature and 100 ml of water was added. After filtration, the residue was washed with water several times to afford the oxidized PGNF in black powder form. In an ice bath, 25 ml of $H_2SO_4$ was stirred, and the oxidized PGNF was added. To the solution, 1.0 g of potassium permanganate was slowly added. The mixture was stirred at 35°C for 8 h. Thereafter, 100 ml of water was added to the reaction mixture, followed by cooling to 0°C; afterward, 5 ml of 30% hydrogen peroxide was slowly added. The obtained dark yellowish solution was centrifuged. The precipitate was washed three times with 3.4% hydrochloric acid and acetone. After drying, a black powder was obtained. A portion (100 mg) of the product was sonicated in 100 ml of water for 2 h, after which it was centrifuged for 20 min at 10 000 rpm. The supernatant was collected as the nGO dispersion. The result of CHN elementary analysis is as follows: H: 2.3(%), C: 45.2(%), N: 0.4(%), O: 52.1(%).

## 2.4. Preparation of three-dimensional GO network in the w/o emulsion system

Ten milligrams of the nGO was dispersed in a solution of 4 ml of water and 1 ml of ethanol. Ethanol was added to facilitate the formation of stable micelles [24,25]. EDA (0.11 g, 1.8 mmol) or OBEA (0.18 g, 1.7 mmol) was added as the cross-linking agent with N-hydroxy succinimide (NHS; 0.058 g; 0.5 mmol) and 1-ethyl-3-(3-dimethylaminopropyl) carbodiimide (EDC; 0.096 g; 0.5 mmol). The solution was added dropwise to 50 ml of cyclohexane with 2.8 g of Span 80. The solution was emulsified by sonication, after which it was stored in the dark overnight. Thereafter, it was washed with acetone and water three times and subsequently dried. The schematic of the synthesis is shown in electronic supplementary material, figure S1 [26].

## 2.5. Adsorption and release experiment

In 20 ml of an aqueous methyl orange solution (25 mg l$^{-1}$), 5 mg of nGO, nGO-EDA and nGO-OBEA were dispersed, separately. After the dispersions were concentrated, water was added to bring the volumes back to the initial 20 ml of methyl orange. The supernatant was collected by centrifugation, and the methyl orange concentration was measured by ultraviolet–visible (UV–Vis) spectroscopy. The schematic adsorption and release processes are described in electronic supplementary material, figure S2. The adsorption rates of the nGO and GO networks were correlated to the change in the methyl orange concentration from the initial concentration (25 mg l$^{-1}$ of methyl orange). The released methyl orange amount was estimated from the difference in the methyl orange concentration before and after stimuli, such as heating (50 or 80°C), ultrasonic wave or near-infrared light (NIR; 650 nm, 40 mW cm$^{-2}$).

# 3. Results and discussion

Using 50 nm diameter PGNFs as the starting material, nGOs with sizes less than 50 nm were synthesized. For PGNF, graphene sheets at 0.3 nm intervals were observed by HR-TEM, as shown in figure 1a,b. For the nGO obtained by oxidation and exfoliation of PGNF, sheets with a diameter of about 10 nm or less were observed, and a hexagonal lattice of benzene rings could be confirmed, as shown in figure 1c,d. As shown in the electronic supplementary material, figure S3, the thickness was approximately 0.8 nm by AFM and XRD measurements, which corresponds to previously reported GO thicknesses [13]. In addition, the presence of oxygen groups such as hydroxyl, epoxide, carbonyl and carboxyl groups was confirmed by FTIR and XPS. The results of elemental analysis from the XPS C1 spectrum and CHN elemental analyser are shown in electronic supplementary material, tables S1 and S2. The C/O ratio from CHN elemental analysis was estimated to be 0.87, which indicates that nGO was sufficiently oxidized.

The three-dimensional GO network structures were prepared by cross-linking carboxyl groups at the edge of nGOs. Using EDA and OBEA, two types of three-dimensional cross-linked nGO structures were obtained, denoted as nGO-EDA and nGO-OBEA, respectively. The formation of an amide group is confirmed in the FTIR spectra, shown in figure 2. The C–H stretching vibration band at 2900 cm$^{-1}$, C–N stretching vibration band at 1100 cm$^{-1}$, and N–H stretching vibration band at 1570 cm$^{-1}$ correspond to the formation of amide cross-linking bridges; meanwhile, the intensities of the C=O stretching vibration band at 1740 cm$^{-1}$, C–OH stretching vibration band at 1210 cm$^{-1}$ and C–O stretching vibration band at 1020 cm$^{-1}$ indicate the consumption of carboxyl groups.

To control the size of the three-dimensional GO network structures, the reaction space was restricted by a w/o emulsion system, where the size of the water droplets was estimated to be 200 nm by DLS, as shown in figure 3b. The resulting three-dimensional network structures appeared to be assembled by nanosheets as shown in SEM and TEM images in figure 4. The image in figure 4 is not very clear. But, when morphologically observed by TEM, the cross-linked structures of nGO-EDA and nGO-OBEA looked like nanosheets assembled at low density rather than dense masses, unlike general stacking and aggregating two-dimensional GOs. The size of three-dimensional GO network structures was not uniform. The approximate size of nGO-EDA was estimated to be approximately 200 nm with a thickness of approximately 10 nm. The approximate size of nGO-OBEA was estimated to be 500 nm with a thickness of several tens of nanometres. The size and thickness of nGO-OBEA are about twice those of nGO-EDA. For nGO-EDA, many nGO sheets with one or a few GO layers were also observed by AFM. These differences between nGO-EDA and nGO-OBEA are assumed to be due to

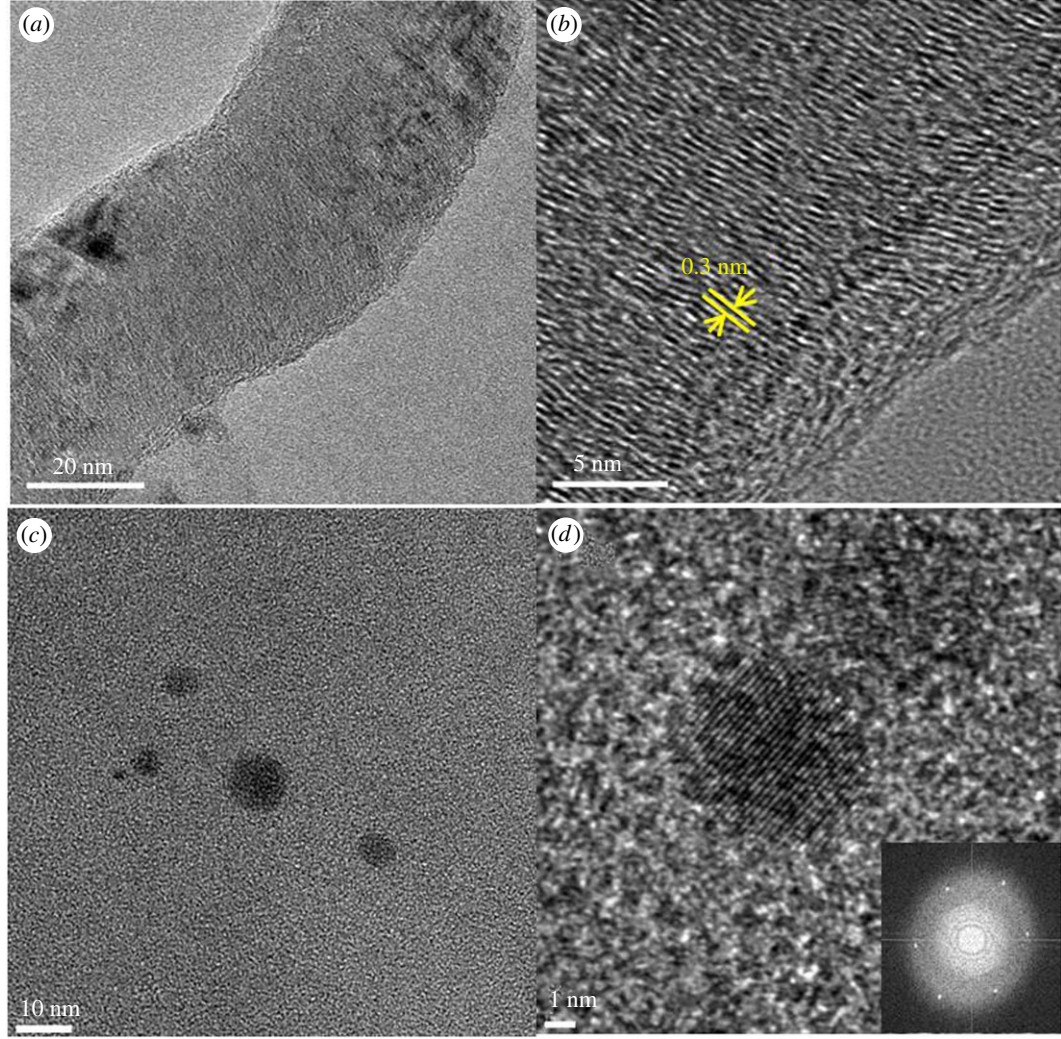

**Figure 1.** HR-TEM images of PGNF (*a,b*) and nGO (*c,d*).

the differences in the reactivity of the cross-linker. The lengths of the linkage of EDA and OBEA were estimated to be 0.38 nm and 0.72 nm, respectively, by molecular calculation between nitrogen atoms, as shown in figure 3*b*. The reactivity of OBEA is higher than that of EDA, because the steric hindrance of the relative long flexible linkages is suppressed. Furthermore, the flat conformation of nGO-EDA and nGO-OBEA is considered to be due to the occurrence of cross-linking reaction at the nGO edges. This is because the carboxyl groups are mainly located at the GO edges or defects rather than at the GO basal plane.

The adsorption and release behaviours of nGO, nGO-EDA and nGO-OBEA were examined by the change in the absorbance of methyl orange solutions. Figure 5 shows the absorption spectra of the supernatants after the methyl orange adsorption on nGO, nGO-EDA and nGO-OBEA adsorbents. Methyl orange solution showed a maximum absorbance at 462 nm. The absorbances at 462 nm were 0.43, 0.36 and 0.24 for nGO, nGO-EDA and nGO-OBEA, respectively. The adsorption amount of methyl orange on nGO, nGO-EDA and nGO-OBEA were calculated to be 10.0, 8.9 and 14.2 mg $l^{-1}$, respectively, corresponding to the change in absorbance at 462 nm. For the supernatant obtained after adsorption on nGO, the absorbance of methyl orange was estimated to be 0.33 from the peak deconvolution results because nGO was dispersed in the supernatant solution. The absorption band at a short wavelength below 300 nm is assigned to the nGOs dispersed in water, as shown in electronic supplementary material, figure S4. The methyl orange loading capacities of nGO, nGO-EDA and nGO-OBEA were estimated to be 40, 36 and 57 mg $g^{-1}$, respectively. These results are summarized in table 1.

The adsorption rate of nGO-EDA is lower than that of nGO. This result may be because the surface area of nGO-EDA including the internal space is almost the same as that of nGO, but the internal space is small. On the other hand, the nGO-OBEA structure with long cross-linking has an adsorption rate

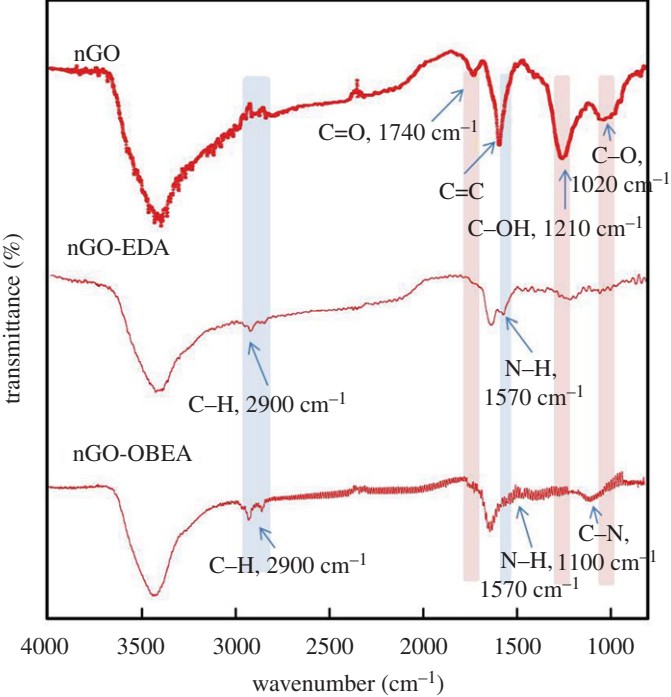

**Figure 2.** FTIR of nGO, nGO-EDA and nGO-OBEA.

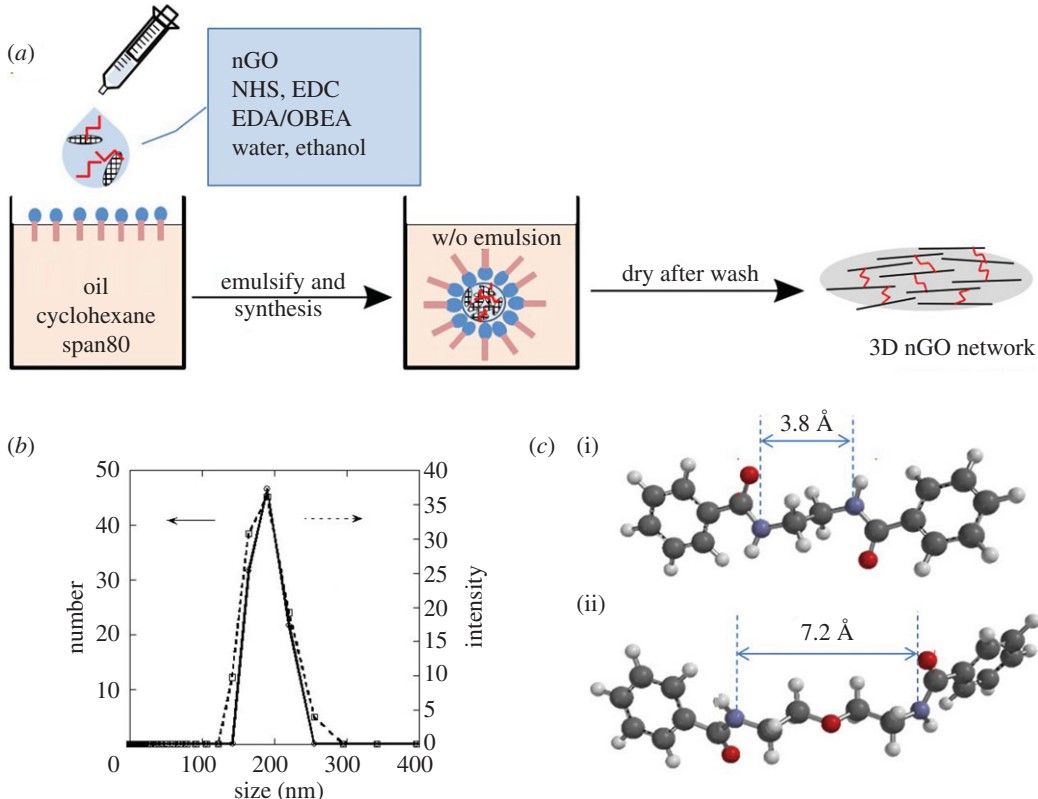

**Figure 3.** Synthetic schemes of nGO-EDA and nGO-OBEA (*a*). DLS measurement of water droplets of w/o emulsion (*b*), where solid line is for number and broken line is for intensity. Calculation results of the cross-link lengths for nGO-EDA (i) and nGO-OBEA (ii) by Spartan '14 (*c*). The nGO structure is simplified and represented by one benzene group.

improved 1.4 and 1.6 times compared to nGO and nGO-EDA. For nGO-OBEA, the internal space is large, and methyl orange molecules may be trapped in the internal space in addition to being adsorbed on the surface of a three-dimensional structure.

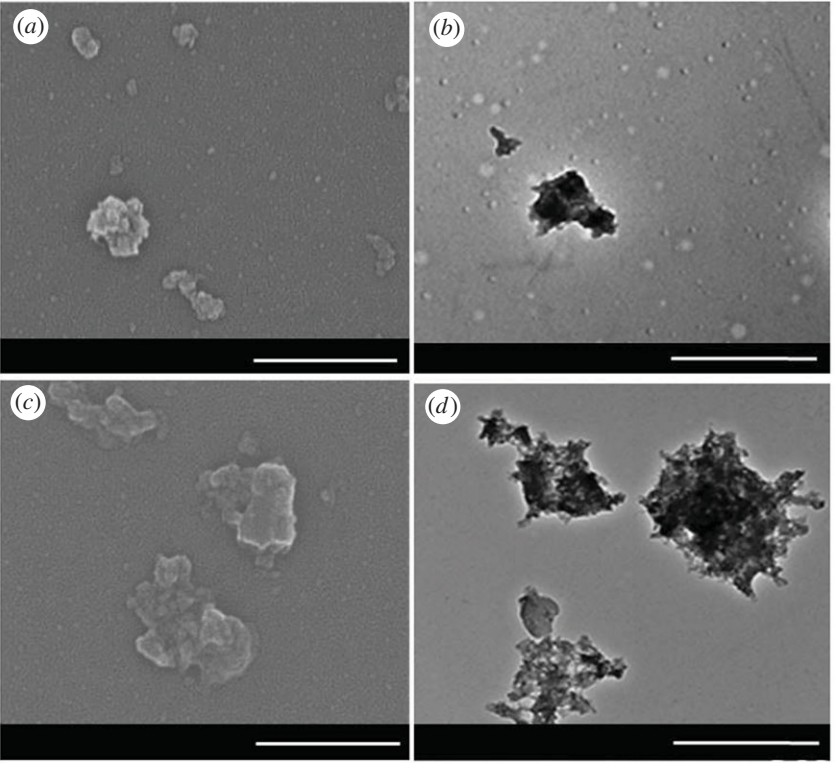

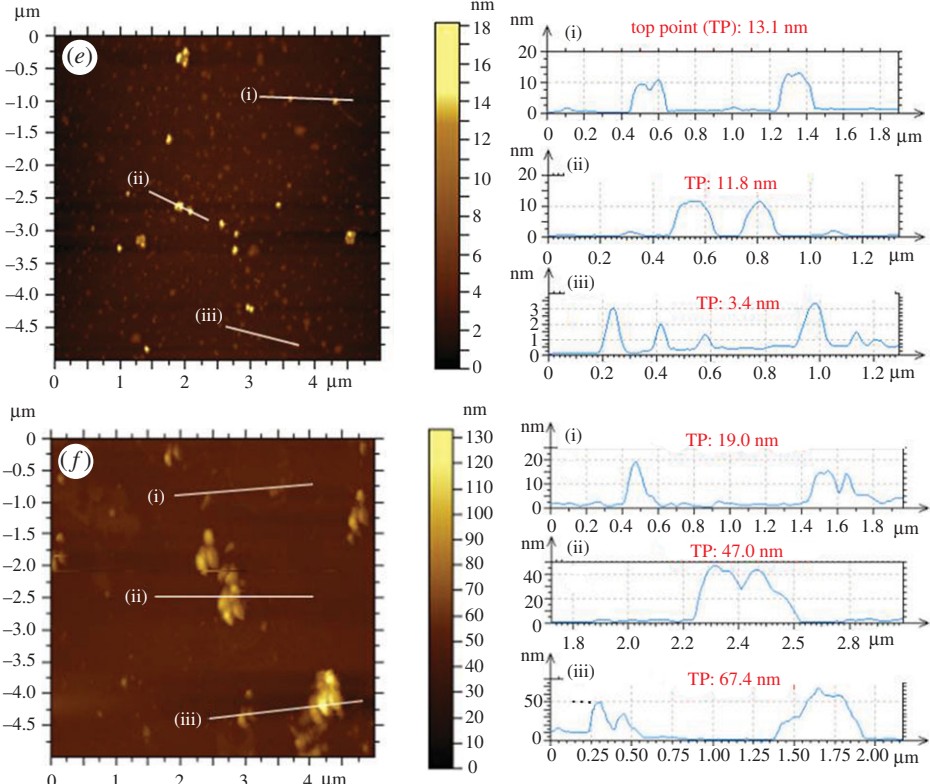

**Figure 4.** SEM and TEM images of nGO-EDA (*a,b*) and nGO-OBEA (*c,d*), scale bar is 500 nm; AFM images of nGO-EDA (*e*) and nGO-OBEA (*f*).

The release rates of methyl orange from nGO, nGO-EDA and nGO-OBEA, under external stimuli, were calculated by UV–vis spectroscopy, and the results were plotted over time, as shown in figure 6. For nGO, the release of the adsorbed methyl orange was considerably challenging; only the

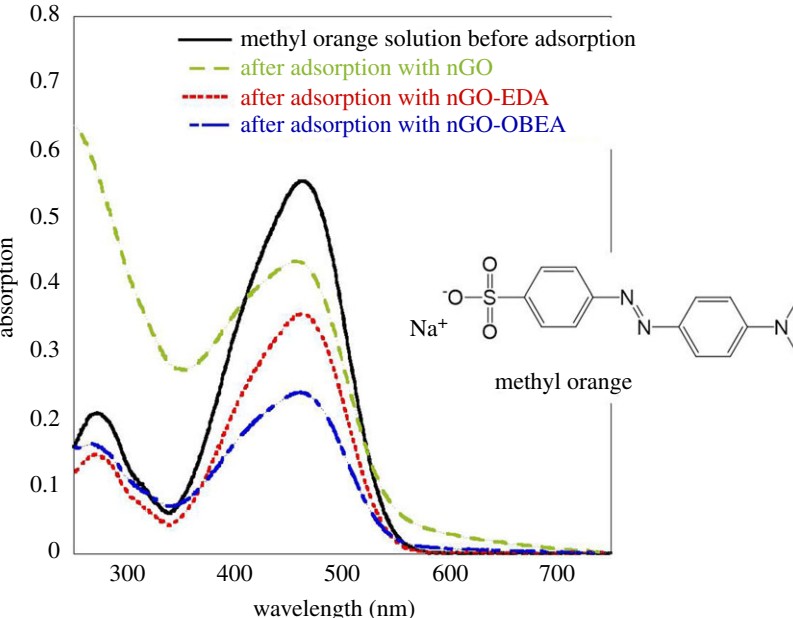

**Figure 5.** Absorption spectra of methyl orange solution before and after the addition of nGO, nGO-EDA and nGO-OBEA. Molecular structure of methyl orange.

**Table 1.** Summarized adsorption amounts of methyl orange on nGO, nGO-EDA and nGO-OBEA.

| methyl orange solution (25 mg l⁻¹) | additives (5 mg) | absorbance of supernatant at 460 nm | adsorption amount (mg l⁻¹) | ratio of adsorption (%) and loading capacity of nGO (mg g⁻¹) |
|---|---|---|---|---|
| 20 ml | — | 0.55 | — | — |
| | nGO | 0.43 (0.33)[a] | 10.0 | 40 |
| | nGO-EDA | 0.36 | 8.9 | 36 |
| | nGO-OBEA | 0.24 | 14.2 | 57 |

[a]Is estimated by peak deconvolution.

aggregated nGO was dispersed by ultrasonication and NIR irradiation, as evidenced by the increase of the absorption band at shorter wavelengths below 300 nm compared to the change in the peak at 462 nm, as shown in electronic supplementary material, figure S5. Aggregation of nGO may have been accelerated by the adsorption of methyl orange, and the methyl orange molecules would be tightly trapped between nGO. By contrast, for nGO-EDA and nGO-OBEA, the peak at 462 nm apparently increased after ultrasonication, NIR irradiation and heat treatment as shown in electronic supplementary material, figures S5 and S6.

Upon ultrasonication, the release rates for both nGO-EDA and nGO-OBEA were the highest, compared to the rates under other stimuli, such as NIR and thermal treatment. By applying ultrasonic waves for only 5 min, the release rates were estimated to be 21% for nGO-EDA and 36% for nGO-OBEA, which were similar to or higher than the results obtained under extended application of other stimuli, such as NIR and thermal treatment at 50°C for 2 h.

For nGO-OBEA, the release behaviour after heating at 80°C was almost the same as that for ultrasonic wave treatment. The high-temperature heat treatment increased the release rate as expected. However, for nGO-EDA, the release rate of methyl orange, upon heating to 80°C for 2 h, was 27%. This value was higher than that obtained in the case of heating to 50°C (17%), but much lower than that for ultrasonic treatment (63%). Regarding this difference in the release rates between nGO-EDA and nGO-OBEA, it is considered that nGO-EDA, which has a short cross linkage, has a small mesh and a narrow internal space; thus, the molecular motion of methyl orange molecules is suppressed. On the other hand, for nGO-OBEA with longer cross linkage, extra methyl orange that does not directly interact with the nGO plane can also be trapped in the internal space by hydrophobic interaction, so

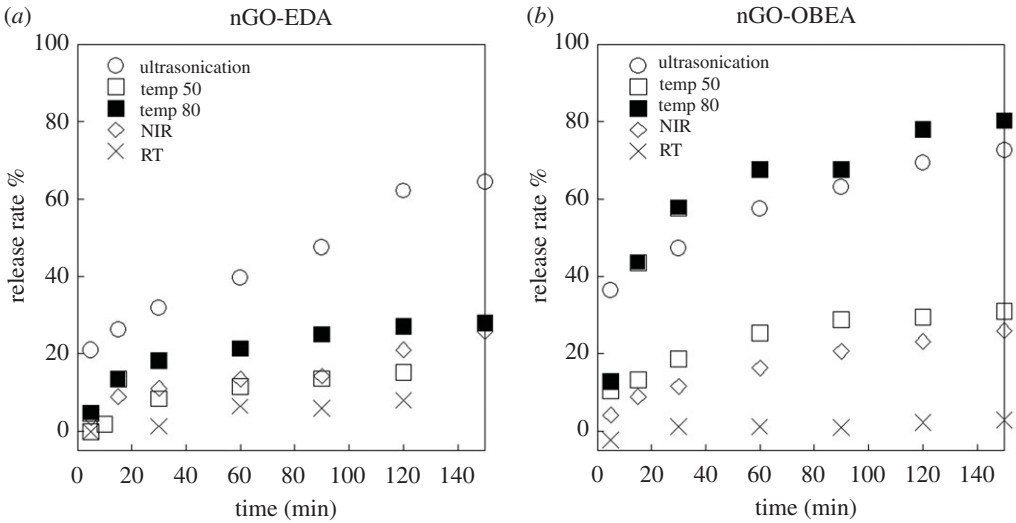

**Figure 6.** Release rate of methyl orange adsorbed on the nGO-EDA (*a*), and nGO-OBEA (*b*) under external stimuli, such as heating at 50℃ (○), heating at 80℃ (△), ultrasonic treatment (□), NIR irradiation (■) and storing in the dark without stimuli(x).

it is easier to release by stimulation. In addition, upon application of ultrasonic waves, the release efficiency might be increased by physical effects, such as cavitation, in addition to molecular motion. For NIR (650 nm) irradiation, there was a slight increase in the solution temperature, as shown in electronic supplementary material, figure S7. This is considered to be due to the generation of molecular vibration energies transferred from the energy absorbed in the aromatic conjugated region of the GO basal plane [27–29]. Although the NIR irradiation of nGO-EDA and nGO-OBEA increased the temperature to 1.3°C and 1.7°C, respectively, the methyl orange release rates were close to the results obtained with heating at 50°C. It is assumed that the π–π interaction between network structures and methyl orange molecules is attenuated by the forced molecular motion under stimuli, so that methyl orange desorbs from the nGO structures.

## 4. Conclusion

Nano-sized three-dimensional GO network structures of different sizes and and cross-linkage lengths were prepared in an emulsion system. In the case of nGO, the release of the adsorbed methyl orange was considerably challenging, whereas three-dimensional network structures of nGO-EDA and nGO-OBEA, which has an internal space effective for the molecular motions under stimuli, successfully adsorbed and released methyl orange. The ultrasonication was the most effective stimulus for methyl orange release on three-dimensional GO networks. The methyl orange adsorption and release rates on nGO-OBEA were higher than those on nGO-EDA. The different length of cross-linkage resulted in a different size and structure of the network. Since the cross-linkage of nGO-OBEA is longer than that of nGO-EDA, there is a less steric hindrance to the synthetic reaction, possibly resulting in a larger architecture of lager meshes.

Data accessibility. The datasets supporting this article have been uploaded as part of the electronic supplementary material. The supplementary materials are available at the Dryad Digital Repository (https://doi.org/10.5061/dryad.0rxwdbrxh) [26].

Authors' contributions. S.K. carried out the acquisition of data and drafted the manuscript. S.M. and S.M. carried out the synthesis and characterization of materials. T.O. participated in data analysis. T.F. gave guidance to the research and helped draft the manuscript. S.K. conceived of and designed the study. All authors gave final approval for publication.

Competing interests. There are no conflicts to declare.

Funding. We acknowledge JSPS KAKENHI grant no. 18K05238 and The Ogasawara Foundation for the Promotion of Science & Engineering. This work was partly supported by JSPS KAKENHI grant nos. JP19H02692, JP18K05238 and JP16H06506 in Scientific Research on Innovative Areas 'Nano-Material Optical-Manipulation'.

Acknowledgements. This work was also supported by Adaptable and Seamless Technology transfer Program through Target-driven R&D (A-STEP) from Japan Science and Technology Agency (JST), The Research Program of 'Five-star Alliance' in 'NJRC Mater. & Dev'.

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
