## [Peer Review File · Royal Society Open Science]

Review History

RSOS-201585.R0 (Original submission)

Review form: Reviewer 1

Is the manuscript scientifically sound in its present form?

No

Are the interpretations and conclusions justified by the results?

Yes

Is the language acceptable?

Yes

Do you have any ethical concerns with this paper?

No

Have you any concerns about statistical analyses in this paper?

No

Recommendation?

Major revision is needed (please make suggestions in comments)

Comments to the Author(s)

The work presented by Kim et al. on the 3D GO network structures prepared by cross-linking nano-sized GOs. Their methyl orange adsorption and release behaviors were investigated under external stimuli, such as thermal treatment, ultrasonic wave treatment, and near-infrared light irradiation. The 3D network structures having internal space effective for the molecular motions under stimuli successfully adsorbed and released methyl orange. After careful review, I suggest major revision subjected to the corrections for the publication in Royal Society Open Science. Some reviewer's comments are listed below.

- (1) The description in the introduction, such as "Graphene, which was recently discovered." Graphene was discovered a long time back in 2004; therefore, the above description should be modified.
- (2) In Figure 3, synthetic schemes of nGO-EDA and nGO-OBEA should be labeled as Figure 3a, DLS measurement as Figure 3b, and so on.
- (3) The results of nGO, nGO-EDA, and nGO-OBEA should be compared with the conventional GO to demonstrate the advantage of nGO in this work.
- (4) The authors should add Raman data to better understand and explain EDA and OBEA's cross-linkage on nGO.
- (5) The authors should add XRD and XPS for nGO-EDA and nGO-OBEA for comparison.
- (6) There are some minor errors throughout the manuscript, such as "Synthestic schemes." The authors are recommended to check the manuscript carefully.
- (7) The authors describe in Figure 4 as "3D network structures appeared to be assembled by nanosheets," as shown in SEM and TEM images. However, 3D network structures appeared to be agglomerated. What do the authors think? Further, SEM and TEM images in Figure 4 are not clear. The authors are recommended to add more convincing images.
- (8) The adsorption amount of methyl orange on nGO, nGO-EDA, and nGO-OBEA was calculated to be 10.0, 8.9, and 14.2 mg/L, respectively, corresponding to the change in absorbance at 462 nm. The methyl orange loading capacities of nGO, nGO-EDA, and nGO-OBEA were estimated to be 40, 36, and 57 mg/g, respectively. The nGO-OBEA structure with a long cross-linkage length (0.72 nm) showed an improved adsorption rate than nGO and nGO-EDA. However, why does nGO-EDA show a decreased adsorption amount of methyl orange and adsorption rate than pure nGO? The authors should explain this phenomenon in detail.

Review form: Reviewer 2

Is the manuscript scientifically sound in its present form?

Yes

Are the interpretations and conclusions justified by the results?

Yes

Is the language acceptable?

Yes

Do you have any ethical concerns with this paper?

Yes

Have you any concerns about statistical analyses in this paper?

No

Recommendation?

Accept with minor revision (please list in comments)

Comments to the Author(s)

This paper present the preparation and characterization of 3D network architectures constructed using nano-sized graphene oxide (nGO) as the building block and demonstrated for adsorption and release of MO. This is well written paper adapting reported synesthetic concepts to fabricate some novel 3d structure. However results are not outstanding results and achievable by other materials more efficiently. Paper can considered wit publication with few points for improvements.

1. Authors should provide some more characterization data (Raman and XRD) to confirm fabrication their nGO.
2. Authors should provide some comparative results with other porous materials.
3. Authors hold clarify what are goals of their designed delivery system and expected kinetics? Release results showed very fast release of 2 hours which is not exciting and much useful as many materials will show the same.

Decision letter (RSOS-201585.R0)

Dear Dr Kim:

Title: Adsorption and release on three-dimensional graphene oxide network structures
Manuscript ID: RSOS-201585

Thank you for submitting the above manuscript to Royal Society Open Science. On behalf of the Editors and the Royal Society of Chemistry, I am pleased to inform you that your manuscript will be accepted for publication in Royal Society Open Science subject to minor revision in accordance with the referee suggestions. Please find the reviewers' comments at the end of this email. I apologise for the delay in sending you this decision.

The reviewers and handling editors have recommended publication, but also suggest some minor revisions to your manuscript. Therefore, I invite you to respond to the comments and revise your manuscript.

Because the schedule for publication is very tight, it is a condition of publication that you submit the revised version of your manuscript before 13-Feb-2021. Please note that the revision deadline will expire at 00.00am on this date. If you do not think you will be able to meet this date please let me know immediately.

Kind regards,
Dr Laura Smith
Publishing Editor, Journals

On behalf of the Subject Editor Professor Anthony Stace and the Associate Editor Professor Tobias Hertel.

RSC Associate Editor:
Comments to the Author:
(There are no comments.)

RSC Subject Editor:
 Comments to the Author:
 (There are no comments.)

Reviewer comments to Author:
 Reviewer: 1

Comments to the Author(s)

The work presented by Kim et al. on the 3D GO network structures prepared by cross-linking nano-sized GOs. Their methyl orange adsorption and release behaviors were investigated under external stimuli, such as thermal treatment, ultrasonic wave treatment, and near-infrared light irradiation. The 3D network structures having internal space effective for the molecular motions under stimuli successfully adsorbed and released methyl orange. After careful review, I suggest major revision subjected to the corrections for the publication in Royal Society Open Science. Some reviewer's comments are listed below.

- (1) The description in the introduction, such as "Graphene, which was recently discovered." Graphene was discovered a long time back in 2004; therefore, the above description should be modified.
- (2) In Figure 3, synthetic schemes of nGO-EDA and nGO-OBEA should be labeled as Figure 3a, DLS measurement as Figure 3b, and so on.
- (3) The results of nGO, nGO-EDA, and nGO-OBEA should be compared with the conventional GO to demonstrate the advantage of nGO in this work.
- (4) The authors should add Raman data to better understand and explain EDA and OBEA's cross-linkage on nGO.
- (5) The authors should add XRD and XPS for nGO-EDA and nGO-OBEA for comparison.
- (6) There are some minor errors throughout the manuscript, such as "Synthestic schemes." The authors are recommended to check the manuscript carefully.
- (7) The authors describe in Figure 4 as "3D network structures appeared to be assembled by nanosheets," as shown in SEM and TEM images. However, 3D network structures appeared to be agglomerated. What do the authors think? Further, SEM and TEM images in Figure 4 are not clear. The authors are recommended to add more convincing images.
- (8) The adsorption amount of methyl orange on nGO, nGO-EDA, and nGO-OBEA was calculated to be 10.0, 8.9, and 14.2 mg/L, respectively, corresponding to the change in absorbance at 462 nm. The methyl orange loading capacities of nGO, nGO-EDA, and nGO-OBEA were estimated to be 40, 36, and 57 mg/g, respectively. The nGO-OBEA structure with a long cross-linkage length (0.72 nm) showed an improved adsorption rate than nGO and nGO-EDA. However, why does nGO-EDA show a decreased adsorption amount of methyl orange and adsorption rate than pure nGO? The authors should explain this phenomenon in detail.

Reviewer: 2

Comments to the Author(s)

This paper present the preparation and characterization of 3D network architectures constructed using nano-sized graphene oxide (nGO) as the building block and demonstrated for adsorption and release of MO. This is well written paper adapting reported synesthetic concepts to fabricate some novel 3d structure. However results are not outstanding results and achievable by other materials more efficiently. Paper can considered wit publication with few points for improvements.

1. Authors should provide some more characterization data (Raman and XRD) to confirm fabrication their nGO.

2. Authors should provide some comparative results with other porous materials.
3. Authors hold clarify what are goals of their designed delivery system and expected kinetics?
Release results showed very fast release of 2 hours which is not exciting and much useful as many materials will show the same.

Author's Response to Decision Letter for (RSOS-201585.R0)

See Appendix A.

RSOS-201585.R1 (Revision)

Review form: Reviewer 2

Is the manuscript scientifically sound in its present form?

Yes

Are the interpretations and conclusions justified by the results?

Yes

Is the language acceptable?

Yes

Do you have any ethical concerns with this paper?

No

Have you any concerns about statistical analyses in this paper?

No

Recommendation?

Accept as is

Comments to the Author(s)

NA

Review form: Reviewer 3

Is the manuscript scientifically sound in its present form?

Yes

Are the interpretations and conclusions justified by the results?

Yes

Is the language acceptable?

Yes

Do you have any ethical concerns with this paper?

No

Have you any concerns about statistical analyses in this paper?

No

Recommendation?

Accept as is

Comments to the Author(s)

The authors revised the manuscript carefully according to the comments. I recommend for publication.

Decision letter (RSOS-201585.R1)

Dear Dr Kim:

Title: Adsorption and release on three-dimensional graphene oxide network structures
Manuscript ID: RSOS-201585.R1

It is a pleasure to accept your manuscript in its current form for publication in Royal Society Open Science. The chemistry content of Royal Society Open Science is published in collaboration with the Royal Society of Chemistry.

On behalf of the Subject Editor Professor Anthony Stace and the Associate Editor Professor Tobias Hertel.

RSC Associate Editor:
Comments to the Author:
(There are no comments.)

RSC Associate Editor:
Comments to the Author:
(There are no comments.)

Reviewer(s)' Comments to Author:
Reviewer: 2

Comments to the Author(s)
NA

Reviewer: 3

Comments to the Author(s)
The authors revised the manuscript carefully according to the comments. I recommend for publication.

Appendix A

Dear Dr Laura Smith
Publishing Editor, Journals

Title: Adsorption and release on three-dimensional graphene oxide network structures
Manuscript ID: RSOS-201585

Thank you for your mail, and I am very happy with this decision. The manuscript has been revised in response to the comments of the reviewers. We are also grateful that we have found improvements and improve the quality of my manuscript. We really hope that it will be published in the RSC Advances, published by the Royal Society of Chemistry. Please kindly consider the revised manuscript again.

I attached the reports include a point by point response to the reviewers' comments and highlighted the changes in the manuscript.

Yours sincerely,
Sunnam Kim

Comments to the Author(s)

The work presented by Kim et al. on the 3D GO network structures prepared by cross-linking nano-sized GOs. Their methyl orange adsorption and release behaviors were investigated under external stimuli, such as thermal treatment, ultrasonic wave treatment, and near-infrared light irradiation. The 3D network structures having internal space effective for the molecular motions under stimuli successfully adsorbed and released methyl orange. After careful review, I suggest major revision subjected to the corrections for the publication in Royal Society Open Science.

Some reviewer's comments are listed below.

(1) The description in the introduction, such as "Graphene, which was recently discovered." Graphene was discovered a long time back in 2004; therefore, the above description should be modified.

→ The sentence of "Graphene, which was recently discovered." was deleted and revised.

"Graphene, comprising a single layer of sp²-hybridized carbon atoms arranged in a hexagonal lattice, has attracted significant interest throughout the scientific community."

(2) In Figure 3, synthetic schemes of nGO-EDA and nGO-OBEA should be labeled as

Figure 3a, DLS measurement as Figure 3b, and so on.

→ As reviewer's comments, the synthetic scheme is labeled as Figure 3a. and the labels of DLS and calculation results are also revised to figure 3b and figure 3c, respectively.

(3) The results of nGO, nGO-EDA, and nGO-OB EA should be compared with the conventional GO to demonstrate the advantage of nGO in this work.

→ We agree with the reviewer's opinion that the comparative research of the DDS properties of these nGO structures should be performed using conventional GO, to address the advantage of size effect. But, the nGO is basically GO, except that its size is limited to nano-order. Conventional GOs are typically manufactured in a wide range of sizes, from a few nanometers to a size of 10 micron. In this study, we aimed to prepare nanosized GO structures for drug carriers based on research reports that a drug carrier size of around 100 nm is appropriate. Their adsorption and release behavior of nGO and nGO networks were investigated as comparison between 2D and 3D structure. In the future, we would like to do a comparative study on the DDS effect of nGO structures with that of GO.

(4) The authors should add Raman data to better understand and explain EDA and OB EA's cross-linkage on nGO.

→ To demonstrate the cross-linking structures, the formation of amide group was confirmed by FTIR spectra. FTIR data provided more information on functional group than Raman, although both Raman spectrum and the infrared spectrum are vibration spectra based on the vibration of the molecule. Since the vibration mode based on the amide group having C=O bond and C-N bond due to the cross-linking reaction appears stronger in the infrared spectrum than in the Raman spectrum. In addition, we did not expect significant peak appearance or the change on D and G bands assigned to sp³ and sp² domains of nGO structures by cross-linking reaction. Please understand the missing Raman data.

(5) The authors should add XRD and XPS for nGO-EDA and nGO-OB EA for comparison.

→ nGO-EDA and nGO-OB EA which differ only in the cross-linking length are considered to be amorphous networks. Therefore it is considered that their 3D structures could not be explained by XPS and XRD data because they are not crystalline or have periodic space. Please understand the missing XPS and XRD data. However, we added the discussion of the structural comparison in relation to the adsorption and release behavior in the text as follows;

“The adsorption rate of nGO-EDA is lower than that of nGO. This result may be because the surface area of nGO-EDA including the internal space is almost the same as that of nGO, but the internal space is small. On the other hand, the nGO-OBEA structure with long cross-linking has an adsorption rate improved 1.4 and 1.6 times compared to nGO and nGO-EDA. For nGO-OBEA, the internal space is large, and methyl orange molecules may be trapped in the internal space in addition to being adsorbed on the surface of 3D structure.”

(6) There are some minor errors throughout the manuscript, such as “Synthestic schemes.” The authors are recommended to check the manuscript carefully.

→ I am very sorry. I'll carefully check the manuscript. The error of “Synthestic schemes.” is revised to “synthetic scheme” in the caption of Fig.3.

(7) The authors describe in Figure 4 as “3D network structures appeared to be assembled by nanosheets,” as shown in SEM and TEM images. However, 3D network structures appeared to be agglomerated. What do the authors think? Further, SEM and TEM images in Figure 4 are not clear. The authors are recommended to add more convincing images.

→ Generally, 2D structure is easily stacked and aggregated but GO is dispersed in the water by oxygen groups on the surface. Although nGO may be reduced by cross linking reaction and they could be easily aggregated, we believe 3D GO structures are not dense cluster but network structure with internal space. Because, crosslinked structures of nGO-EDA and nGO-OBEA looked like nanosheets assembled at low density rather than dense masses, when their morphology is observed by TEM. It is also demonstrated with the adsorption rate results depending on the structure. We added the additional explain in the text.

“The image in Figure 4 is not very clear. But, when morphologically observed by TEM, the cross-linked structures of nGO-EDA and nGO-OBEA looked like nanosheets assembled at low density rather than dense masses, unlike general stacking and aggregating 2D GOs.”

As mentioned in response to (5), we added the discussion of the structural comparison in relation to the adsorption and release behavior as follows;

“The adsorption rate of nGO-EDA is lower than that of nGO. This result may be because the surface area of nGO-EDA including the internal space is almost the same as that of nGO, but the internal space is small. On the other hand, the nGO-OBEA structure with long cross-linking has an adsorption rate improved 1.4 and 1.6 times compared to nGO and nGO-EDA. For nGO-OBEA, the internal space is large, and methyl

orange molecules may be trapped in the internal space in addition to being adsorbed on the surface of 3D structure.”

(8) The adsorption amount of methyl orange on nGO, nGO-EDA, and nGO-OB EA was calculated to be 10.0, 8.9, and 14.2 mg/L, respectively, corresponding to the change in absorbance at 462 nm. The methyl orange loading capacities of nGO, nGO-EDA, and nGO-OB EA were estimated to be 40, 36, and 57 mg/g, respectively. The nGO-OB EA structure with a long cross-linkage length (0.72 nm) showed an improved adsorption rate than nGO and nGO-EDA. However, why does nGO-EDA show a decreased adsorption amount of methyl orange and adsorption rate than pure nGO? The authors should explain this phenomenon in detail.

→ Absolutely we agreed to review’s comments and we added additional explain in the text as mentioned response to (5). For nGO-EDA with short and rigid cross-linking, the internal space is expected to be small.

Reviewer: 2

Comments to the Author(s)

This paper present the preparation and characterization of 3D network architectures constructed using nano-sized graphene oxide (nGO) as the building block and demonstrated for adsorption and release of MO. This is well written paper adapting reported synesthetic concepts to fabricate some novel 3d structure. However results are not outstanding results and achievable by other materials more efficiently. Paper can considered wit publication with few points for improvements.

1.Authors should provide some more characterization data (Raman and XRD) to confirm fabrication their nGO.

→ nGO-EDA and nGO-OB EA which differ only in the cross-linking length are considered to be amorphous networks. Therefore it is considered that their 3D structures could not be explained by XPS and XRD data because they are not crystalline or have periodic space. Please understand the missing XPS and XRD data. However, we added the discussion of the structural comparison in relation to the adsorption and release behavior in the text as follows;

“The image in Figure 4 is not very clear. But, when morphologically observed by TEM, the cross-linked structures of nGO-EDA and nGO-OB EA looked like nanosheets assembled at low density rather than dense masses, unlike general stacking and aggregating 2D GOs.”

“The adsorption rate of nGO-EDA is lower than that of nGO. This result may be

because the surface area of nGO-EDA including the internal space is almost the same as that of nGO, but the internal space is small. On the other hand, the nGO-OBEA structure with long cross-linking has an adsorption rate improved 1.4 and 1.6 times compared to nGO and nGO-EDA. For nGO-OBEA, the internal space is large, and methyl orange molecules may be trapped in the internal space in addition to being adsorbed on the surface of 3D structure.”

To demonstrate the cross-linking structures, the formation of amide group was confirmed by FTIR spectra. FTIR data provided more information on functional group than Raman, although both Raman spectrum and the infrared spectrum are vibration spectra based on the vibration of the molecule. Since the vibration mode based on the amide group having C=O bond and C-N bond due to the cross-linking reaction appears stronger in the infrared spectrum than in the Raman spectrum. In addition, we did not expect significant peak appearance or the change on D and G bands assigned to sp³ and sp² domains of nGO structures by cross-linking reaction. Please understand the missing Raman data.

On the other hand, nGO-EDA and nGO-OBEA which differ only in the cross-linking length are considered to be amorphous networks. Therefore it is considered that their 3D structures could not be explained by XRD data because they are not crystalline or have periodic space. Please understand the missing XRD data.

2. Authors should provide some comparative results with other porous materials.

→ Silica gel is well known as one of porous materials. According to the report by Suman Koner et al, silica gel coated with cationic surfactant of CTAB has an absorption capacity of 36mg/g, which is less than nGO-OBEA. (Ref: Suman Koner et al, International Journal of Current Research, Vol. 3, Issue, 6, pp.128-133, June, 2011) Looking at the values alone, our results are better than those reported by Suman Koner.

On the other hand, adsorption capacity depends on surface area, pore size, surface charge, etc., so it would be difficult to make a direct comparison. Methyl orange (MO) is an anionic dye, although it has a hydrophobic part. Since the GO surface has hydroxy and carboxy groups, it is considered that ionic repulsion acts between GO and MO, weakening adsorption capacity. Although not described in this paper, when an adsorption experiment using DOX instead of MO revealed that the adsorption rate was 90% or more.

3. Authors hold clarify what are goals of their designed delivery system and expected

kinetics? Release results showed very fast release of 2 hours which is not exciting and much useful as many materials will show the same

→ We aim to develop a carrier that releases drugs when stimulated. There was almost no release when no stimulus was given. As intended, release behavior was confirmed only by stimulation. On the other hand, in the case of DOX, although not described in this paper, the adsorption rate was high, but DOX was not released even when stimulated. For DOX, very stable adsorption properties were observed. It turns out that these properties are strongly dependent on the drug molecule.